



# In situ observations of greenhouse gases over Europe during the CoMet 1.0 campaign aboard the HALO aircraft

Michał Gałkowski[1,2], Armin Jordan[1], Michael Rothe[1], Julia Marshall[1], Frank-Thomas Koch[1,3], Jinxuan Chen[1], Anna Agusti-Panareda[4], Andreas Fix[5], and Christoph Gerbig[1]

[1]Department of Biogeochemical Systems, Max Planck Institute for Biogeochemistry, Jena, Germany
[2]Faculty of Physics and Applied Computer Science, AGH University of Science and Technology, Kraków, Poland
[3]Meteorological Observatory Hohenpeissenberg, Deutscher Wetterdienst, Germany
[4]European Centre for Medium-Range Weather Forecasts, Reading, UK
[5]Deutsches Zentrum für Luft- und Raumfahrt (DLR), Institut für Physik der Atmosphäre, Oberpfaffenhofen, Germany

**Correspondence:** Michał Gałkowski (michal.galkowski@bgc-jena.mpg.de)

**Abstract.**

The intensive measurement campaign CoMet 1.0 (Carbon dioxide and Methane mission) took place during May and June 2018, with a focus on greenhouse gases over Europe. CoMet 1.0 aimed at characterising the distribution of $CH_4$ and $CO_2$ over significant regional sources with the use of a fleet of research aircraft, as well as validating remote sensing measurements from state-of-the-art instrumentation installed on-board against a set of independent in-situ observations. Here we present the results of over 55 hours of accurate and precise in situ measurements of $CO_2$, $CH_4$ and CO mixing ratios made during CoMet 1.0 flights with a cavity ring-down spectrometer aboard the German research aircraft HALO, together with results from analyses of 96 discrete air samples collected aboard the same platform. A careful in-flight calibration strategy together with post-flight quality assessment made it possible to determine both the single measurement precision as well as biases against respective WMO scales. We compare the result of greenhouse gas observations against two of the available global modelling systems, namely Jena CarboScope and CAMS (Copernicus Atmosphere Monitoring Service). We find overall good agreement between the global models and the observed mixing ratios in the free-tropospheric range, characterised by very low bias values for the CAMS $CH_4$ and the CarboScope $CO_2$ products, with a mean free tropospheric offset of 0 (14) ppb and 0.8 (1.3) ppm respectively, with the quoted number giving the standard uncertainty in the final digits for the numerical value. Higher bias is observed for CAMS $CO_2$ (equal to 3.7 (1.5) ppm), and for CO the model-observation mismatch is variable with height (with offset equal to -1.0 (8.8) ppb). We also present laboratory analyses of air samples collected throughout the flights, which include information on the isotopic composition of $CH_4$, and we demonstrate the potential of simultaneously measuring $\delta^{13}C-CH_4$ and $\delta^2H-CH_4$ from air to determine the sources of enhanced methane signals using even a limited amount of discrete samples. Using flasks collected during two flights over the Upper Silesian Coal Basin (USCB, southern Poland), one of the strongest methane-emitting regions in the European Union, we were able to use the Miller-Tans approach to derive the isotopic signature of the measured source, with values of $\delta^2H$ equal to -224.7 (6.6) ‰ and $\delta^{13}C$ to -50.9 (1.1) ‰, giving significantly lower $\delta^2H$ values compared to previous studies in the area.



## 1 Introduction

Increased mixing ratios of atmospheric greenhouse gases (GHGs) are recognised as the primary cause of the warming observed
in the climate system over the past 70 years. Of these, the most important are carbon dioxide ($CO_2$) and methane ($CH_4$),
respectively responsible for approximately 56 % and 32 % of the globally-averaged increase in radiative forcing caused by
greenhouse gases, as compared to the pre-industrial period (IPCC et al., 2013). Further increases in the atmospheric burden
of greenhouse gases are expected to lead to a multitude of negative impacts over a wide range of climate system components
throughout the 21$^{st}$ century and beyond. These include further temperature increase, sea level rise, changes in precipitation
patterns, shrinking of ice cover and more. Furthermore, cumulative emissions of $CO_2$ will have lasting effects on most aspects
of climate for many centuries, even if anthropogenic emissions are stopped altogether (IPCC et al., 2013).

The accuracy of climate projections is substantially reduced, however, by uncertainties in the specific components of green-
house gas budgets, which stem either from difficulties in precise estimation of direct sinks and emissions, or from our limited
understanding of specific feedback processes. Despite the critical importance of this issue, our knowledge about even the two
most important anthropogenically-influenced greenhouse gases, $CO_2$ and $CH_4$, is still inadequate. In fact, even though intense
scientific and political activities have targeted this area of research over the past 20 years, the uncertainties related to the most
important source and sink processes remain high (Ballantyne et al., 2015), reflecting the enormous complexity of the Earth
System, with its multitude of elements and feedback mechanisms, operating on a vast range of spatial and temporal scales.

Main sources of uncertainties in the reported budgets are similar for both $CO_2$ and $CH_4$. When considering bottom-up
methods, they are related either to a) the lack of representativeness of flux measurement sites used for up-scaling the fluxes
from specific source areas, or b) incomplete knowledge at the process level, which affects the emission models used for the
calculation of either emission factors or actual fluxes. Top-down methods, in turn, are based on inverse modelling and critically
depend on the availability of high-precision atmospheric observations in the areas studied, which is still insufficient. In order
to significantly reduce the global uncertainties in the budgets of greenhouse gases using ground-based instrumentation, a
significant expansion of the observation networks is required to provide precise regional budgets for the most important source
and sink areas (Ciais et al., 2014). Observation networks of sufficient density are currently only available over Europe and parts
of North America, where they have been used successfully to constrain anthropogenic and biogenic fluxes of greenhouse gases
(e.g. Bergamaschi et al., 2018).

Utilising space-borne observations can bridge the data gap by providing high-resolution data on regional scales across
the globe, which has driven significant developments in remote sensing techniques since the mid-1990s. Since the launch of
SCIAMACHY in 2002, remote sensing data on global distributions of column-averaged dry-air mixing ratios for atmospheric
carbon dioxide ($XCO_2$) and methane ($XCH_4$) comes most often from surface-reflected near- and short-wave infrared radiation
detectors (Bovensmann et al., 1999; Kuze et al., 2009; Reuter et al., 2011; Butz et al., 2012; Eldering et al., 2012; Reuter
et al., 2019). While important insights into greenhouse gas budgets have been gained (Bergamaschi et al., 2013; Basu et al.,
2013), there are still significant limitations when using infrared methods (Kirschke et al., 2013; Le Quéré et al., 2018). As an
alternative to passive remote sensing, the Integrated Path Differential Absorption (IPDA) technique has been adapted in recent



years to provide column-averaged measurements of greenhouse gas mixing ratios with high accuracy (e.g. Amediek et al., 2008; Sakaizawa et al., 2009; Spiers et al., 2011; Dobler et al., 2013; Du et al., 2017; Amediek et al., 2017). All of these remote sensing techniques rely heavily on the availability of independent calibration and validation data sets.

Aircraft measurements are flexible and constitute a critical link for bridging the gap between ground-based networks and space-borne observations in constraining emissions at multiple scales. They can be performed either with precise in situ measurement techniques that can be calibrated and made traceable to WMO calibration scales (e.g., Wofsy, 2011; Sweeney et al., 2015; Filges et al., 2018; Boschetti et al., 2018; Umezawa et al., 2018) or utilizing remote sensing instruments (Krings et al., 2013). Airborne observations can be applied to describe regional and local variability of the observed signals (Wofsy, 2011;

Sweeney et al., 2015). They can also be used as validation of the coupled transport-emission models (Ahmadov et al., 2007; Sarrat et al., 2007; Park et al., 2018; Leifer et al., 2018), or used to directly infer the fluxes of measured components. Such direct inference has been demonstrated in the past, e.g. using Gaussian plume models (Krings et al., 2013), Lagrangian mass-balance approaches (Karion et al., 2013; Cambaliza et al., 2014) or regional Bayesian inverse-modelling systems (e.g.: Saeki et al., 2013; Boschetti et al., 2018). Other techniques of flux estimation also exist (c.f. Varon et al., 2018).

In order to further push the limits and improve the observation and modelling methods developed in the past, a multi-platform aircraft research mission was envisaged, designed, proposed and executed in collaboration between the German Aerospace Center (DLR), the Max Planck Institute for Biogeochemistry (MPI-BGC), the University of Bremen, the Free University of Berlin, AGH University of Science and Technology, and other partners. CoMet 1.0 (Carbon Dioxide and Methane Mission, see the overview paper by Fix et al. in this Special Issue), executed in May and June 2018, targeted hot-spots of $CO_2$ and $CH_4$

emissions in Europe, with a strong focus on the Upper-Silesian Coal Basin in Poland, one of the largest regional emitters of methane. The mission utilised a multitude of state-of-the art instruments applied on both airborne and ground based platforms, including active lidar (CHARM-F, see Amediek et al., 2017), passive remote sensing (MAMAP, see Gerilowski et al., 2011; Krings et al., 2013), in situ measurements (CRDS, QCLS; see e.g. Filges et al., 2018), and satellite observations. Wherever possible, these were applied simultaneously in order to a) achieve high observation inter-comparability, b) test the limits of

applied measurement techniques, c) provide a rich suite of observations to evaluate atmospheric transport models, and d) to estimate regional GHG emissions.

Here we present the results of in situ observations of atmospheric greenhouse gases and methane isotopic composition obtained over nine research flights of the German research aircraft HALO during CoMet 1.0, with the use of two airborne instruments installed aboard the aircraft during the campaign: i) JIG (Jena Instrument for Greenhouse gas measurements), a

continuous analyser for measurements of $CH_4$, CO, $CO_2$ and $H_2O$ and ii) JAS (Jena Air Sampler), which collected discrete $1\,l$ samples for subsequent laboratory analyses of $CH_4$, $CO_2$, $H_2$, $N_2O$, $SF_6$, $O_2/N_2$, $Ar/N_2$, $\delta^{13}C-CH_4$ and $\delta^2H-CH_4$.





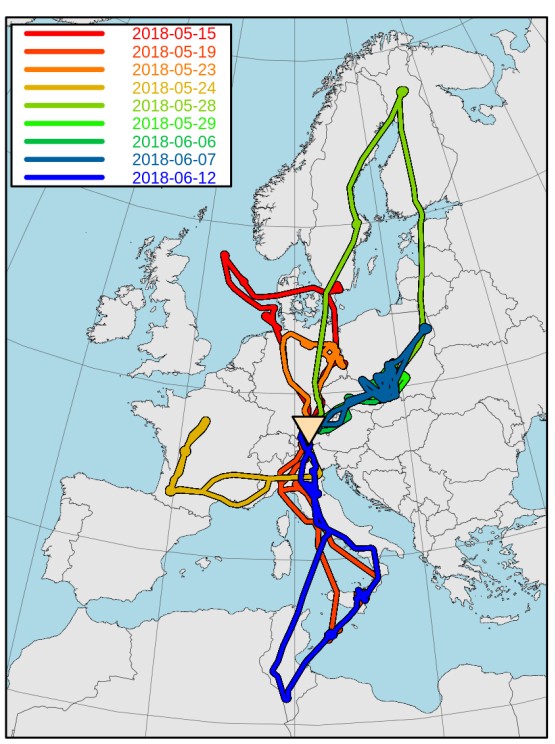

**Figure 1.** Geographical extent of HALO research flights during the CoMet 1.0 mission.

## 2 Methods

### 2.1 CoMet 1.0 flights

During the CoMet 1.0 mission, HALO performed nine research flights, with more than 63 hours of observations over continental Europe and parts of northern Africa (Fig. 1), with the base of operations located in Oberpfaffenhofen (Bavaria, Germany, marked with a triangle in the figure). During the campaign, each flight aimed to reach several scientific goals based on the synoptic meteorological conditions over selected target areas. These goals included, e.g., comparisons between active remote sensing and in situ observations, co-located measurements at satellite overpass points, comparisons against another airborne instrument, and others. For each of those, a specific measurement strategy was adopted. Those relevant for the measurements discussed in this study are described in section 2.3. A complete description of the CoMet 1.0 mission will be given in an overview publication by Fix et al. (in prep, this special issue).



## 2.2 In situ instrumentation

### 2.2.1 JIG - Jena Instrument for Greenhouse gas measurements

In situ continuous airborne measurements of greenhouse gases on board HALO have been carried out using JIG (Jena Instrument for Greenhouse gas measurements; photo available in the supplement, Fig. S1). The core of the device is a modified commercial analyser G4201-m, developed by Picarro Inc. (Santa Clara, CA, USA), which was redesigned in order to fulfil conditions necessary for long-term deployment in the scope of IAGOS ERI (In-service Aircraft for a Global Observing System - European Research Infrastructure). Detailed development of the instrument, with the description of its operational parameters, is described in Chen et al. (2010) and Filges et al. (2015, 2018). Here, only the basic operation principle and main differences to the IAGOS setup are given.

The core method of the measurement is wavelength-scanning Cavity Ring-Down Spectroscopy (CRDS), where an infrared-wavelength laser light is injected into a high finesse optical cavity. In the first phase of the measurement, the strength of the incident laser beam gradually increases over time thanks to the resonance effect in the optical cavity, which also allows for the enhancement of the effective absorption length and thus increases the detector's sensitivity. After reaching the designated signal level, the laser is turned off and the ring-down phase of the measurement begins. The time constant of the resulting exponential decay (ring-down time) depends on the absorption coefficient of the measured compound for the laser wavelength, tuned so that the scan along selected individual spectral lines of the measured molecules is possible. The measurement requires the usage of calibration gases, as well as careful control over cavity pressure and temperatures in order to prevent sample density variations. During the measurements described here, the cavity pressure was set at all times to 140.0 Torr, and the temperature to 45.00 °C, with tollerance levels of 0.1 Torr and 0.02 °C, respectively.

Calibration of the instrument was performed in the laboratory before and after the CoMet 1.0 mission by measuring three air mixtures, stored in so-called working tanks, which covered the range of ambient mixing ratios of $CO_2$, $CH_4$ and CO and had assignments traceable to the respective WMO calibration scales. All JIG trace gas mixing ratio data provided in the current study are reported on the current WMO calibration scales: $CO_2$ X2007, $CH_4$ X2004A, and CO X2014A. The instrument calibration was monitored during the mission with the use of two reference in-flight cylinders containing dry mixtures of atmospheric air of known composition, which were analysed several times during each flight. The calibration cycle consisted of two intervals, each five minutes in length. The initial two minutes of each interval were discarded in subsequent analyses due to pressure equilibration effects within the regulators.

Except for a single calibration check performed prior to take-off during a power-up procedure, all the other calibration check cycles were enabled manually by an on-board operator of the system, during transit phases of the flight, in contrast to the regular IAGOS implementation (Filges et al., 2015, 2018). The in-flight calibration checks occurred at high altitudes, where high gradients of GHGs were not expected and the loss of information could be minimised. The last calibration cycle was always performed immediately before the final approach of the flight. It should be noted that the results of the in-flight calibration checks were only used for assessing a potential drift in the instrument calibration factors relative to the pre-mission (April 2018) and post-mission (November 2018) laboratory calibrations.



Additional, independent verification of the measurement quality was carried out by comparison of the mixing ratios from in situ measurements and those obtained from laboratory analyses of air samples collected by the JAS (Jena Air Sampler, see sec. 2.2.2, and the discussion of the results in section 3.1).

Two malfunctions of the JIG occurred during the CoMet 1.0 mission. On May 28[th] (flight no. 5), a software issue (i.e. clock
reset) caused the loss of 96.8 % of in situ data from that day. The remaining 3.2 % have been excluded from the following analysis due to their fragmentation. The second malfunction occurred during the power-up procedure on June 7[th], 2018 (flight no. 8), when JIG suffered an unexpected shutdown due to cabin overheating, which required a manual reset of a temperature switch. This in turn caused unintentional damage to the optical fibre mount located inside the instrument housing. The resulting loss of signal strength caused deterioration of the system parameters over flights no. 8 and 9, increasing noise and shifting the
instrument calibration parameters. These were subsequently corrected using in-flight calibrations and post-mission laboratory calibrations. The impact of the malfunction and final effect of applied corrections is discussed in section 3.1.

### 2.2.2 JAS - Jena Air Sampler

The sampler used during CoMet 1.0 (Fig. S1) is an airborne version of the automated flask sampler developed within the ICOS (Integrated Carbon Observing System) infrastructure. The device is equipped with 12 slots for holding one litre glass flasks
with automatically operated valves at both ends. Sample air, collected outside the aircraft fuselage with a dedicated inlet, flows through tubing (PFA, 415 cm, 1/4″ OD) and into the drying unit (70 $cm^3$, SS) filled with magnesium perchlorate. The dryer is connected via another tubing section (PFA, 317 cm, 1/4″ OD, plus additional 15 cm of 1/4″ OD, SS, for pressure sensor mount) to a Teflon diaphragm pump (N 813.3, KNF Neuberger GmbH) that provides the over-pressure necessary to flush and pressurise the flasks. The pump is connected directly to the main input manifold (passing through all three rack-mounted
sub-units) via another flexible tube (PFA, 156 cm, 1/4″ OD). The input manifold can be connected to the output line either via open flasks (one or more), or a two-way bypass valve. At the end of the sampler flow line, a mass flow meter (MFM) (D6F-20A6-000, Omron) is installed that integrates the total volume of air flowing through an opened flask, which ensures that the flask volume has been sufficiently flushed with the sampled air (at least 6 l under normal conditions). At the outlet of the system, a pressure release valve is installed that prevents the back-flow of the pressurised cabin air into the sampler in case of
power failure. Three pressure sensors and three thermometers are also installed to monitor the status of the system.

The sampler is controlled via computer in the electronic control section using dedicated software. The procedure for flask flushing and filling is enabled manually by an operator, either at pre-determined flight altitudes (in the case of measurements of vertical profiles) or locations (e.g. plume sampling).

In order to precisely establish spatio-temporal coordinates from which the sample is collected, a flow model has been used
that takes into account i) flow information from the MFM, ii) the volume of tubing elements such as the dryer and tubing, and iii) the varying physical length of tubing between the inlet and the flask inlet slots (ranging from 10.76 m to 15.58 m). For each collected sample, a temporal weighting function was calculated that represents the collected air volume, following the approach suggested by Chen et al. (2012).





| Compound | Precision | Uncertainty of scale link | Unit |
|---|---|---|---|
| $CO_2$[a] | 0.065 | 0.046 | ppm |
| $CH_4$[a] | 1.3 | 0.70 | ppb |
| $N_2O$[a] | 0.13 | 0.12 | ppb |
| $H_2$[a] | 0.31 | 0.28 | ppb |
| $SF_6$[a] | 0.044 | 0.025 | ppt |
| $O_2/N_2$[b] | 1.5 | 1.6 | per meg |
| $Ar/N_2$[b] | 4.5 | 6.0 | per meg |
| $\delta^{13}C-CH_4$[c] | 0.046 | 0.12 | ‰ |
| $\delta^2H-CH_4$[c] | 0.49 | 1.4 | ‰ |

[a] - Precision calculated as the standard error of the repeated flask measurements (usually between three to five). An average standard error for complete set of flasks collected during CoMet 1.0 is given. Uncertainty of the scale link specified as root from sum of squared uncertainties of: i) specified CCL (Central Calibration Laboratory) scale transfer uncertainties, ii) precision limit of individual laboratory standard calibration events, and iii) response drifts between successive calibration events. For $H_2$, scale transfer uncertainty is equal to zero by definition, as flasks were measured directly against the primary scale. This uncertainty estimate does not include the accuracy of the respective WMO scale.

[b] - $O_2/N_2$ and $Ar/N_2$ measurements were done on the BGC-IsoLab local realisation of the Scripps scale. Realisation is achieved through the regular measurements of in-house standards against independently calibrated tanks from Scripps. Reproducibility estimate is given as the average of standard deviations calculated from measurements against in-house standards (IsoLab, MPI-BGC, Jena) of three cylinders with air mixtures calibrated independently at Scripps Institute for Oceanography (SIO), covering the $O_2/N_2$ range between -262.2 per meg to -807.6 per meg and $Ar/N_2$ range of 136.5 per meg to 167.1 per meg.

[c] - Only a single measurement of each sample was possible. Precision estimated using repeated working standard measurements performed in sequence with the sample (usually four or five per sample). Reproducibility defined according to Sperlich et al. (2016).

**Table 1.** Average measurement uncertainties of the flasks collected during CoMet 1.0. All values given with with coverage probability of 0.68. A correction factor based on a Student's t-distribution was applied to account for low population size, following the Guide to the Expression of Uncertainty in Measurement (JCGM, 2008).

All flasks collected aboard HALO during CoMet 1.0 were analysed in the GasLab of the Max Planck Institute for Biogeochemistry (MPI-BGC) in Jena, Germany, to establish mixing ratios of trace gases ($CO_2$, $CH_4$, $N_2O$, $H_2$, $SF_6$) based on their respective WMO scales. Additional analyses of $O_2/N_2$, $Ar/N_2$ and isotopic composition of methane ($\delta^{13}C-CH_4$ and $\delta^2H-CH_4$) were carried out in the IsoLab of MPI-BGC. The typical measurement precision of the laboratory analyses is given in Table 1.



A significant drift in CO mixing ratios was observed over the period between sample collection and the subsequent mea-
surement in the GasLab. The resulting enhancement in the mixing ratio was not easily correctable, therefore the results had to
be discarded. Careful quality control and additional tests did not show any signs of other gases being affected.

## 2.3 Flight patterns

Depending on the scientific goals set out before each research flight, different flight patterns were executed in order to obtain the
most valuable data. The main strategies adopted for the CoMet 1.0 mission were: i) long-range gradient observations, designed
to maximise the amount of observations for active lidar measurements with CHARM-F operated on HALO, ii) vertical profiles,
aimed mainly at the intercomparison between the lidar and in situ observations and iii) low-altitude legs, performed to assess
the enhancements of $CO_2$ and $CH_4$ downwind of their sources (plume-chasing).

### 2.3.1 Large-scale variability in upper troposphere and lower stratosphere

Due to the constraints related to using other instruments (the active lidar), a significant amount of flight time was spent flying
level at altitudes higher than $4\,\mathrm{km}$, in order to emulate a flight path similar to that of a satellite system. Typical variability of in
situ greenhouse gas mixing ratios was low in these cases and is usually considered to be caused by intermixing of air masses
coming from different regional source areas. In situ data obtained in this manner are well-suited for validation of Global
Chemistry Models. As an example, in sec. 3.2 we compared JIG observations against well established modelling products:
CAMS greenhouse gas forecasts. A detailed model description is given in sec. 2.4.

### 2.3.2 Vertical profiles

Multiple vertical profiles of the atmosphere were carried out during the campaign in order to establish the connection between
column-integrated remote sensing and in situ measurements, thus also linking remote sensing observations to common WMO
scales for greenhouse gases. The typical strategy consisted of i) a high-altitude overflight over a selected target, ii) descent in
the form of a spiral to the lowest possible altitude above the target, iii) subsequent ascent back to high altitude, usually flown
along the shortest path in the direction of the next planned way-point.

Usually two or three vertical profiles were executed during a given flight, depending on the availability of points of inter-
est and airspace accessibility. Wherever possible, profiles were executed above a) ICOS stations, b) TCCON stations (Total
Carbon Column Observing Network), c) Sentinel 5P or GOSAT overpass locations. Flasks were also collected during vertical
soundings, at levels distributed between the minimum and maximum altitude, typically consisting of six samples per profile
(in some cases reduced to four).

Measurements of vertical profiles are also of high interest for model validation exercises, as the availability of highly pre-
cise data on greenhouse gases over Europe is currently still limited. In the scope of the current study, we have assessed the
performance of two well-established modelling products (CAMS and Jena CarboScope, see sec. 2.4) against CoMet 1.0 in situ
observations.





### 2.3.3 Measurements in the planetary bounary layer - plume chasing and isotopic composition

A limited amount of data were also collected inside the planetary boundary layer (PBL), usually during the lowest stage of vertical profile sounding. In several cases, however, these PBL sections were extended in order to cross low-level emission plumes (plume chasing). Of particular interest here are the flights over the Upper Silesian Coal Basin (a large regional methane source) and downwind of the Bełchatów coal power plant (the largest single emitter of $CO_2$ in Europe, according to the European Pollutant Release and Transfer Register, v16; E-PRTR, 2019). For both of these sources, clear enhancements from the strong sources were captured when crossing the plume downwind.

Additional to the in situ measurements, flasks were also collected to gather information about additional compounds and the stable isotopic composition of $CH_4$. For the cases where sufficient data were available, we have applied the method of Miller and Tans (2003), a variation of a classic Keeling model (Keeling, 1958), in order to obtain the isotopic mean source signature ($\delta_0$), expressed using relative delta notation. The method assumes a two-factor mixing of background air and methane-enhanced plume:

$$\delta_{obs}\chi_{obs} = \delta_0\chi_{obs} - \chi_{bg}(\delta_{bg} - \delta_0), \tag{1}$$

where $\delta_{obs}$ is the measured isotopic signature, $\delta_{bg}$ is the background signature, $\chi_{obs}$ is the mixing ratio of the analysed compound, and $\chi_{bg}$ is the background mixing ratio. Here, similar to the Keeling approach, information on $\delta_0$ can be gleaned from the application of linear regression, however the source signature is calculated from the slope, rather than intercept, of the linear fit formula. Following the study by Cantrell (2008), we have applied a Williamson-York regression, which allows one to take into account uncorrelated errors in both the X- and Y-axes of the data.

The Miller-Tans method relies on the appropriate assignment of the background signature (i.e. of the atmospheric air outside of the plume). Long term data available from atmospheric observations show that the isotopic composition of methane in the atmosphere is variable (Röckmann et al., 2016; Nisbet et al., 2019) in both space and time. In order to estimate the background signature, here we have used measurements from air samples collected in the immediate vicinity of the target plume, either from i) the upwind air masses when possible or ii) air outside of the main plume when not.

### 2.4 Models

As part of the Copernicus Atmosphere Monitoring Service (CAMS), the European Centre for Medium-Range Weather Forecasts (ECMWF) performs greenhouse gas simulations based on its Integrated Forecasting System (IFS) and provides operational global forecasting products focused on greenhouse gases. In this work, we have used the five-day high resolution greenhouse gas forecast product from CAMS (experiment ID: gqpe, downloaded in April 2020; see Agusti-Panareda et al., 2017; Agustí-Panareda et al., 2019) in order to validate the model using our observations. Further in the text, we will refer to these data as CAMS for simplicity. Satellite data were used for initialization of the forecast, namely TANSO-GOSAT for $CO_2$ and $CH_4$ and additionally MetOp-IASI for $CH_4$ (Massart et al., 2014, 2016). For CO, CAMS operational analysis (Inness et al., 2015, 2019) was used for forecast initialization. Original CAMS 1-day forecast data, available at TCo1279 Gaussian cubic



octahedral grid (equivalent to approximately 9-km horizontal resolution) was interpolated to 0.125° x 0.125°. The frequency of the analysed CAMS data was 3-hourly, and vertical resolution was the regular L137 ECMWF configuration.

Additionally, $CO_2$ data were also compared to the Jena CarboScope product (version s04oc_v4.3, Rödenbeck, 2005), further referred to as CarboScope. While the resolution of the driving CarboScope model output fields is much lower in this case (4° x 5° horizontal), the system benefits from using the fluxes of $CO_2$ optimized using a Bayesian inversion framework. A detailed description of the modelling system is given in Rödenbeck et al. (2003) and Rödenbeck (2005). The transport model TM3, which is used by CarboScope, is described in Heimann and Körner (2003).

## 3 Results & Discussion

### 3.1 Overview and data quality

A total of 55 hours and 17 minutes of high-frequency (1 Hz) observations of $CO_2$, $CH_4$ and CO were obtained aboard HALO in the scope of the CoMet 1.0 campaign. Measurements of $CO_2$ are presented in Fig. 2, and a full overview, including also $CH_4$ and CO is available in the supplement (Fig. S2). Observations were performed at altitudes ranging from approximately 50 m up to 14 km above mean sea level. Data from 51 vertical profiles are available these are listed in the supplement, Table S1), out of which 21 have simultaneous flask measurements. 15 in-flight calibrations were performed, making it possible for the single-measurement precision to be estimated for flights no. 1–7. These were equal to: 0.06 ppm ($CO_2$), 0.3 ppb ($CH_4$) and 3.1 ppb (CO) . Malfunction during the roll-out procedure prior to flight no. 8 caused deterioration in the instrument noise for two subsequent flights (no. 8 and 9), with values of precision increasing to 0.3 ppm, 1.5 ppb and 50 ppb for $CO_2$, $CH_4$ and CO, respectively.

The comparison between flask and in situ measurements is available for all except one flight (no. 5). From the total 96 samples collected and analysed, 84 had simultaneous in situ measurements available from JIG that could be used for a bias assessment. As shown in Fig. 3, the average bias for flights 1–7 was equal to -0.131 (30) ppm for $CO_2$ and -2.93 (32) ppb for $CH_4$, where number in brackets represent standard uncertainty in the final digits quoted for the numerical value. Larger spread when independent measurements are considered (Fig. 2) stems mainly from the imperfect match between the temporal coordinates of the two instruments, which can be considered random and does not cause systematical shift. After the malfunction, i.e. for flights no. 8 and 9, these mean offsets were equal to 0.127 (68) ppm and -0.64 (91) ppb for the respective gases. While the difference of values caused by the broken mounting is statistically significant, the difference is still close to the WMO compatibility goal.

### 3.2 Large Scale Variability

Out of the total amount of observations during CoMet 1.0, 84 % were performed at altitudes above 4 km and are of particular interest for model validation. To demonstrate the utility of the observations to validate model results, as well as to help under-



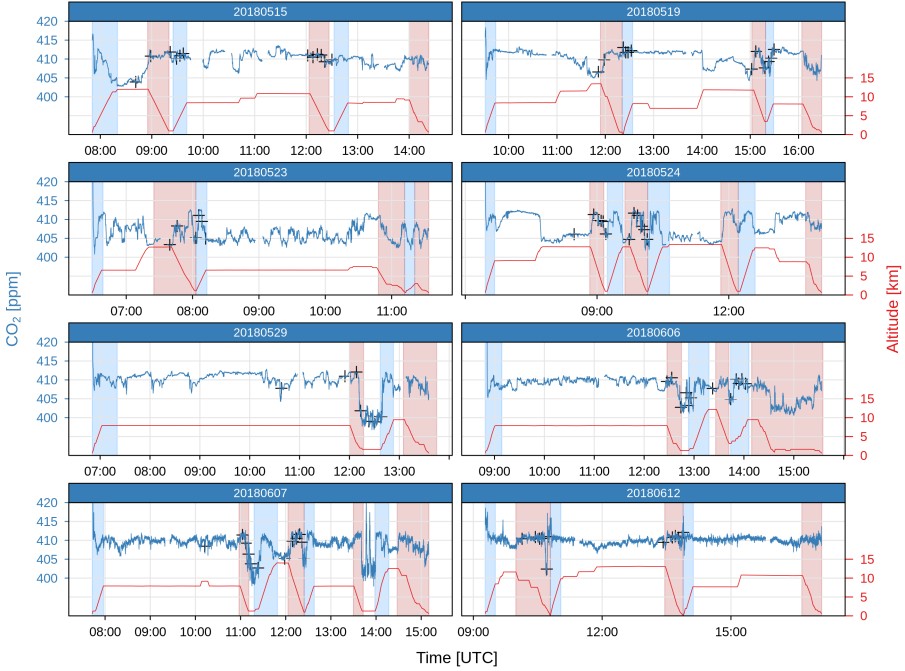

**Figure 2.** In situ mixing ratios of $CO_2$ measured throughout CoMet 1.0 with flight altitudes. Shading corresponds to the vertical profiles discussed throughout the manuscript. Co-located flask measurements are marked with black crosses.

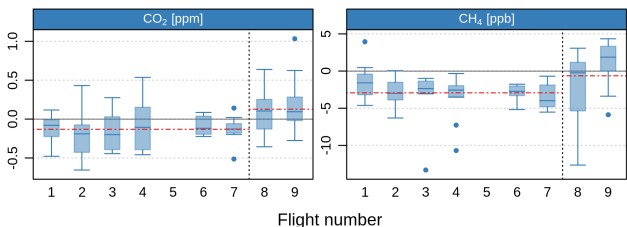

**Figure 3.** Comparison between JIG results and analysis of flasks collected during CoMet 1.0 aboard HALO. Data from the last two flights (separated by the dashed line) indicate a residual change in calibration, following an instrument malfunction. See text for details.

stand the patterns in measured mixing ratios, we analyse and compare JIG measurements to CAMS high-resolution products for $CO_2$, $CH_4$ and CO. Flight no. 2, shown in light red in Fig. 1, is discussed as an example.

The flight (Fig. 4, top) was executed on May 19$^{th}$ 2018, with the main goal of capturing large-scale variability of greenhouse

gases in the atmosphere above Italy and the Mediterranean coast. Two vertical profiles were planned above ICOS stations, namely Lampedusa (35° 31′ 05" N, 12° 37′ 50" E) and the Monte Cimone mountain station (Tuscan-Emilian Apennines, 44° 11′ 38" N, 10° 42′ 05" E). The latter profile was executed approximately 20 km away from the target due to an active





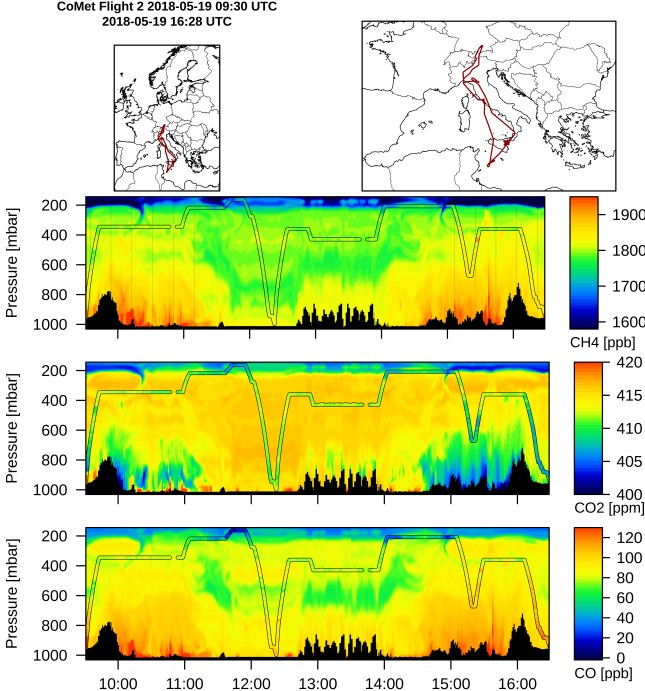

**Figure 4.** Curtain plot showing results for May 19th, 2018. Top row: Overview map with flight path (left), and map zoomed over flight area (right); other rows: time series of aircraft altitude, with CAMS model results as colour plot in background for $CH_4$, $CO_2$, and CO (top to bottom); coloured lines superimposed on the curtain plot denote in situ measurements from HALO, with mixing ratios plotted in the same colour scale.

thunderstorm over the site. Other points of interest were i) the Po valley, crossed twice (morning and afternoon) at high altitude and ii) two high-altitude circles around Mount Etna (35° 31′ 05" N, 12° 37′ 50" E).

Fig. 4, bottom, shows the CAMS model results extracted at the geographical aircraft time and location, together with corresponding in situ observations from JIG overlaid on the aircraft flight path, both plotted using the same colour scale. The model captures most of the features observed in the atmosphere. Speaking in terms of observed spatial and temporal variability of the atmospheric composition (modeled and observed), four sections of the flight can be identified: a) morning overflight over northern Italy, b) passage over Mediterranean Sea, ending with a vertical sounding at Lampedusa, c) circling Mount Etna

at medium altitude and d) northward flight over Italy, including a vertical profile in the vicinity of Monte Cimone, with a subsequent crossing of Po Valley and the Alps, ending with landing at home base.

During the first section of the flight (a), after the initial climb, the aircraft stayed at high altitude (300 hPa, approx. 8 km a.m.s.l.), sounding the free troposphere above the Po Valley. Comparison of measured concentrations to the model suggests that the chosen flight level was well within the free troposphere. Close to the surface, CAMS predicted high enhancements



of greenhouse gases, clearly visible on the $CH_4$ and CO plots. At the time (10:00–11:00 UTC), the boundary layer was still developing, reaching only about 900 hPa level (approximately 1000 m).

After crossing the coastline, the aircraft ascended to a cruise altitude of approx. 12 km. Around 11:30 UTC it reached the tropopause level, and, after another increase in altitude, entered into the stratosphere for approx. 10 minutes immediately prior to the vertical sounding at Lampedusa. The vertical structure of the atmosphere was generally well predicted in the model (see

sec. 3.4, and also Fig. S3-S18 in the supplement), albeit larger differences can be observed in the lowest 3 km of the profile, especially for $CO_2$.

The Mt. Etna section of the flight took place mostly in the free troposphere and no significant gradients were observed for either of the measured compounds. Subsequent transfer over southern Italy (section d) started with an ascent into the stratosphere (at approx. 220 hPa). After approx. 10 minutes of northward flight, the aircraft crossed into the troposphere

horizontally again. Immediately before the descent to Monte Cimone it crossed a stratospheric intrusion, possibly caused by the intensive deep convective mixing occurring in the area in the afternoon on that day. After that the aircraft descended, making a downward spiral over the northern Apennines, down to approx. 3.5 km. The model-observation discrepancy is much higher at this point, most probably due to: i) errors in representation of the local convective systems, or ii) errors in the surface fluxes driving the modelled mixing ratios, or the mixture of both. The high resolution CAMS product correctly captures most

of the large scale phenomena. There are, however, specific situations where the performance of the model drops, specifically in the vicinity of strong local convection systems, where parameterisations can sufficiently predict neither the height nor the transport of the strong enhancements present in the Po valley. The reasons behind this discrepancy may stem from the inability of coarser scale parameterisations to capture local phenomena accurately, or from an incorrect distribution of the ground-level sources.

In the following section, we analyse the model-data mismatch more closely using the subset of CoMet 1.0 data collected only during the vertical soundings.

### 3.3 Vertical structure of the atmosphere

All profiles of $CO_2$, $CH_4$ and CO collected with JIG are presented in Fig. 5, together with comparison to the CarboScope and CAMS model products. Individual comparisons are available in the supplement (Fig. S3-S10 and S11-S18). It should be noted

that the mean profile for the lowest altitudes is dominated by a limited amount of cases when the ground level was reached. This happened most often at home base (EDMO, Oberpfaffenhofen). Similarly, only a limited number of profiles reached altitudes beyond 12 km above mean sea level.

Again, three distinct altitude ranges can be distinguished based on the observed gas mixing ratios and their variance. The lowest, the PBL, is characterised by highly variable concentrations and is located in the altitude range of 0–3 km. Both the

highest and lowest observed concentrations of $CO_2$ were observed here, with most of the observations in the range between 400 and 420 ppm. Occasionally, peaks of over 420 ppm were observed in the vicinity of strong point sources (e.g. Bełchatów power plant). $CH_4$ and CO variability was also high, with most observed values between approximately 1880 and 2000 ppb for methane (with peaks above 2100 ppb) and 100–150 ppb for carbon monoxide.





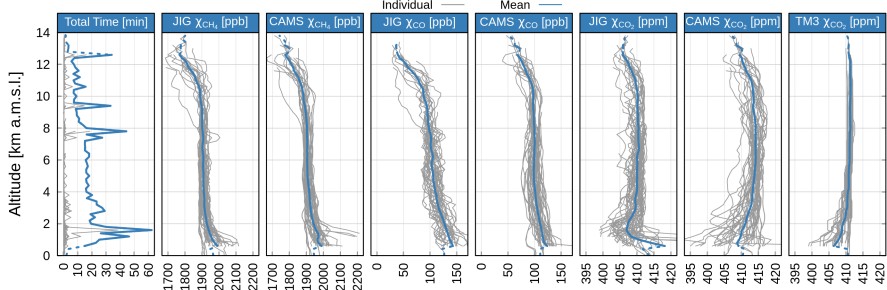

**Figure 5.** An overview of the vertical profiles measured during the CoMet 1.0 mission, together with modelled profiles from CAMS and CarboScope (denoted with TM3). In the first panel, total time is calculated as a sum of 1-s observations from each respective bin. All the other panels present mixing ratios for different variables, binned into 200 m layers. Averages for each layer are shown as a solid blue line while solid grey lines represent individual profiles. Dashed lines represent the means with less than 200 seconds of observations available. Only data from the individual profiles marked on Fig. 2 are plotted here, i.e. the measurements collected during horizontal sections of the flights are not included.

Above the PBL range, free tropospheric observations were characterised by much smaller variability. For $CO_2$ the mean
profile becomes flat with a value of 410 ppm up to approximately 10 km altitude. For $CH_4$ and CO a vertical gradient in the mean values is observed, reflecting the balance between surface anthropogenic sources, large-scale advection and tropospheric chemical sinks.

Above the altitude of 10 km, a more pronounced decrease in the mixing ratios is observed, which is directly related to occasional crossings into the tropopause region and the lowermost stratosphere. The variability of the observed decrease is
large and follows the variability in the tropopause height. On average we have observed a 4 ppm decrease for $CO_2$ between 10 km and 13 km, which is most probably caused the increasing age of the slow-mixing stratospheric air (Andrews et al., 2001). Decreases of $CH_4$ and CO are more pronounced (on average 150 ppb and 70 ppb, i.e. 8 % and 45 % relative to the value at 10 km), underlining an increased oxidative breakdown of these tracers (added to the age effect in case of $CH_4$).

While the observed gradient is similar to previously reported studies (e.g. Wofsy, 2011; Sweeney et al., 2015; Umezawa
et al., 2018), measurements from CoMet 1.0 also clearly indicate the increase in atmospheric concentrations over the past years. For example, the $CH_4$ mixing ratios measured during the IMECC campaign in autumn 2009 (Geibel et al., 2012) were approximately 60 ppb lower throughout the atmospheric column than those observed in 2018. This number is closely in line with mean global atmospheric growth of methane of 63.8 ppb (between 2009 and 2018 NOAA, 2020).

### 3.4 Model validation

The vertical profile subset of the measurements was the basis of the comparison to the well-established global modelling systems CAMS and CarboScope. Here, we focus on describing the vertical structure of the model-data mismatch, defined as the difference between the modelled results and in situ observations from JIG, presented in Fig. 6. Mirroring previous discussion





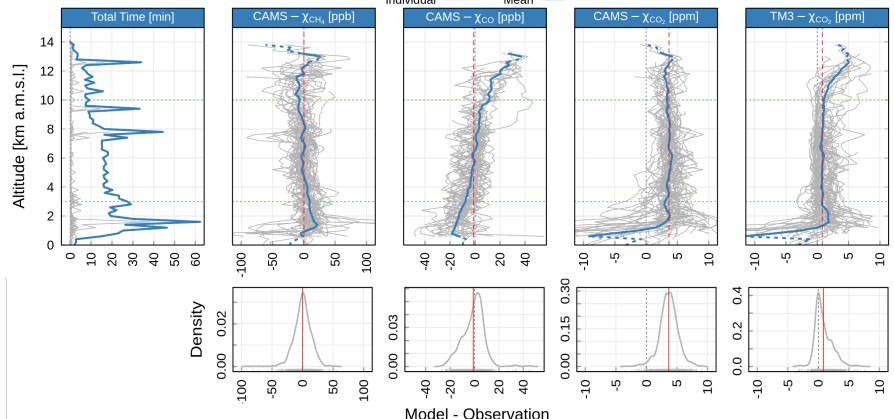

**Figure 6.** Top row: The first panel shows time spent in each altitude bin (200m) for individual flights (grey lines) and the sum for CoMet 1.0 (solid blue). The four panels to the right show differences between modelled (CAMS and CarboScope, here marked as TM3) and measured mixing ratios as a function of altitude (grey - individual flights, blue - average per altitude bin). Dashed red line represents the average value between the altitude range 3–10 km (vertical range marked with horizontal dashed green lines). Bottom: density plots for measurements in the free-tropospheric (3–10 km) range.

of different characteristics of the atmosphere, here the different nature of discrepancies can also be separated into three distinct layers of the atmosphere. It is worth noting that the model-observation mismatch is, in general, constant in neither space nor time, as can be seen when analysing the variability between different flight days. However, some important conclusions can be drawn when analysing the overall vertical structure in the difference between global model results and CoMet 1.0 in situ observations.

The variability in the mismatch is highest closest to the surface (bottom 3 km), which is related to influences from local sources/sinks, as well as variability of atmospheric mixing and transport in the PBL, which are hard to represent at respective model resolutions (0.125° x 0.125° for CAMS, 4° x 5° for CarboScope). Another source of mismatch is related to uncertainties in the emissions data used by the models. Validation of individual emission sources, while of critical importance, remains challenging. In addition, in the case of biospheric $CO_2$, the prediction of fluxes on scales relevant for direct comparison of mixing ratios on regional scales also remains a difficult task. This is true for all the analysed compounds and both models, with a markedly larger discrepancy in the CarboScope product that clearly suffers as a result of its low spatial resolution. As the in situ measurements from CoMet 1.0 are not numerous enough to give a robust estimate for the European region, and differences between the model predictions and observations will be heavily dependent on a specific synoptic range and distribution of sources in the vicinity, we do not provide any general statistics for this lowest part of the atmosphere.

In the free-tropospheric range, the mismatch represents large-scale offset between the model and observations better, and is only weakly dependent on the spatial distribution of the emissions sources. Under this assumption, the mismatch is mostly caused either by i) large (i.e. at least national) scale offsets in emission strengths, ii) bias in the initialisation of the forecasted



fields (with CAMS GHG and operational analysis fields which are a combination of model simulation and satellite observations), iii) errors in chemistry parameterisations (OH radical reaction chains, $CH_4$ and CO).

In the CAMS product, the offset between the modelled values and observations in the troposphere becomes stable with height for $CH_4$ and $CO_2$, with a symmetric distribution around a mean value ($CH_4$: 0 (14) ppb; $CO_2$: 3.7 (1.5) ppm, where standard
uncertainty in the final digits is given in brackets. For $CO_2$, a substantial offset is still present, most probably connected with errors in the strength of the net biospheric fluxes predicted in the model. This general offset needs to be taken into account if the data are either compared directly to the measurements (this manuscript) or used as lateral boundary conditions for regional modelling studies. For CO, a sloped model-data mismatch is observed, most likely related to known issues with the inventories of anthropogenic emissions of CO (e.g. Boschetti et al., 2015) superimposed on chemistry-related effects. The mean value of
the offset of CO in the 3–10 km altitude range is equal to -1.0 (8.8) ppb.

For altitudes above 10 km, the mismatch between CAMS and observations shows larger variability for $CH_4$ and CO, with $CO_2$ discrepancies similar to those observed in the free troposphere. While the number of observations at these higher altitudes is relatively low compared to those below 10 km, we believe that these differences are also caused by errors in both transport and chemistry schemes in the IFS system. These have been investigated in some detail in the case of $CH_4$, for which the errors
in the stratosphere have been found to be larger than those observed in the troposphere (Verma et al., 2017).

Optimised $CO_2$ mixing ratios from CarboScope also show overall good agreement when compared to observations, despite lower model resolution compared to CAMS. The model-data mismatch is dominated by a random term in the free tropospheric range (0.8 (1.3) ppm). Interestingly, the distribution of the mismatch in this altitude range is a positively skewed Gaussian curve (Fig. 6, bottom-right panel), with the main peak almost symmetric around 0 ppm, and the tail being responsible for most
of the offset in the 3–10 km range. The most probable cause is the inability of the model to represent convective uplifting of $CO_2$-depleted air from the PBL. It should also be noted that in the CarboScope product, a systematic over-prediction of $CO_2$ mixing ratios above 10 km (up to 5 ppm) is observed, which might be caused either by i) significant errors in the tropopause height or ii) too fast vertical mixing in the lower stratosphere, leading to underestimation of the gradient and the chemical age of $CO_2$. In the PBL range, the mixing ratios are generally underestimated, sometimes by more than 10 ppm, albeit such
a large discrepancy is only visible for the lowest altitude range (less than 1 km), where the sample size is low. Where the observation set is more robust, the bulk of observations is characterised by discrepancies smaller than 10 ppm and can have either positive or negative sign. Such behaviour is to be expected when trying to compare local plume enhancements to the low-resolution model results that averages over large, inhomogeneous areas characterised by a dynamic spatio-temporal diurnal cycle of fluxes.

## 3.5 Additional data from discrete samples - JAS

Figure 7 presents additional data acquired throughout the campaign with discrete samples, with a detailed overview provided in the supplement (Fig. S19 and Table S1). Apart from $CH_4$ and $CO_2$, for which the flask data were used for validation, important constituents were monitored, offering further insights into the state of the atmosphere over Europe during the CoMet 1.0 mission. The general nature of the collected data follows the patterns described for in situ data, with three distinct abundance





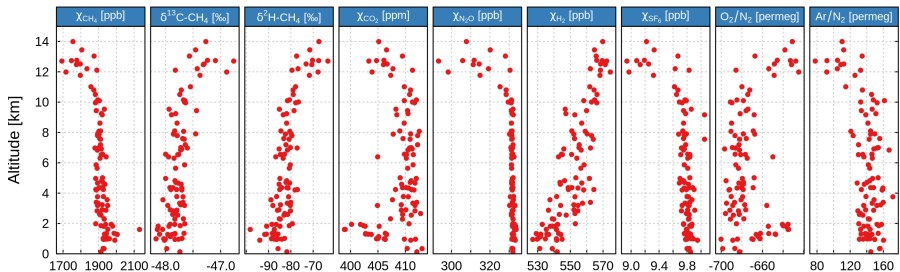

**Figure 7.** Combined results of air composition from flask samples collected during CoMet 1.0 aboard HALO.

regimes: i) PBL, ii) residual layer / free troposphere and iii) tropopause and lower stratosphere, however with some marked differences.

For $N_2O$ and $SF_6$, both potent greenhouse gases (IPCC et al., 2013), there is no clearly visible mixing ratio gradient between the PBL and the free troposphere. For both gases, the variability is known to be dominated by the slow stratospheric transport, effectively causing the "age" of air masses to be higher than the tropospheric air below (Andrews et al., 2001). For $N_2O$, this

effect is superimposed on the additional signal caused by its photo-chemical destruction in the stratosphere. Notably, during CoMet 1.0, two samples were collected with $SF_6$ mixing ratios elevated by approximately 0.2 ppt. The first was filled on June 7[th], at 9.2 km altitude, over Czechia, and second on June 12[th], at 7.6 km, during the downward profile over the Po Valley. The potential source of these two observations might be worth investigating, especially in light of the constant atmospheric increase of the $SF_6$, despite substantial efforts to curb emissions of this potent greenhouse gas (Weiss and Prinn, 2011). Some attention

was also given to molecular hydrogen ($H_2$) due to its potential feedbacks to the atmosphere oxidative capacity and stratospheric ozone levels (see Batenburg et al. (2012) and references therein). Values measured during the mission, namely 540 ppb near the surface, approximately 550–560 ppb throughout the free troposphere and approx. 570 ppb in the lower stratosphere, are comparable to previously reported values, e.g. in the scope of the CARIBIC project (Batenburg et al., 2012). This structure is driven by the presence of a relatively strong soil sink in the latitude band covered during CoMet 1.0, as has been confirmed by

modelling studies (e.g. Pieterse et al., 2011). $O_2/N_2$ and $Ar/N_2$ ratios are presented for completeness, but are not discussed in the present study.

Of particular interest during CoMet 1.0 was the stable isotopic composition of methane. Abundances of both $\delta^{13}C-CH_4$ and $\delta^2H-CH_4$ are strongly and negatively correlated (R = -0.88 and R = -0.96, respectively) with mixing ratios of methane, signifying the potential to use the isotopes as a marker of the source processes. Indeed, in the next section we present an

application of using isotopic composition to differentiate between specific source types in the study area of the USCB.

### 3.6    Capturing the USCB source signature with isotopic data

Due to the broader spatial range covered by the HALO aircraft, the amount of samples taken over the USCB area using the JAS instrument was limited to 12 flasks, collected over two flights performed on May 29[th] and June 6[th], 2018 (HALO flights 6 &



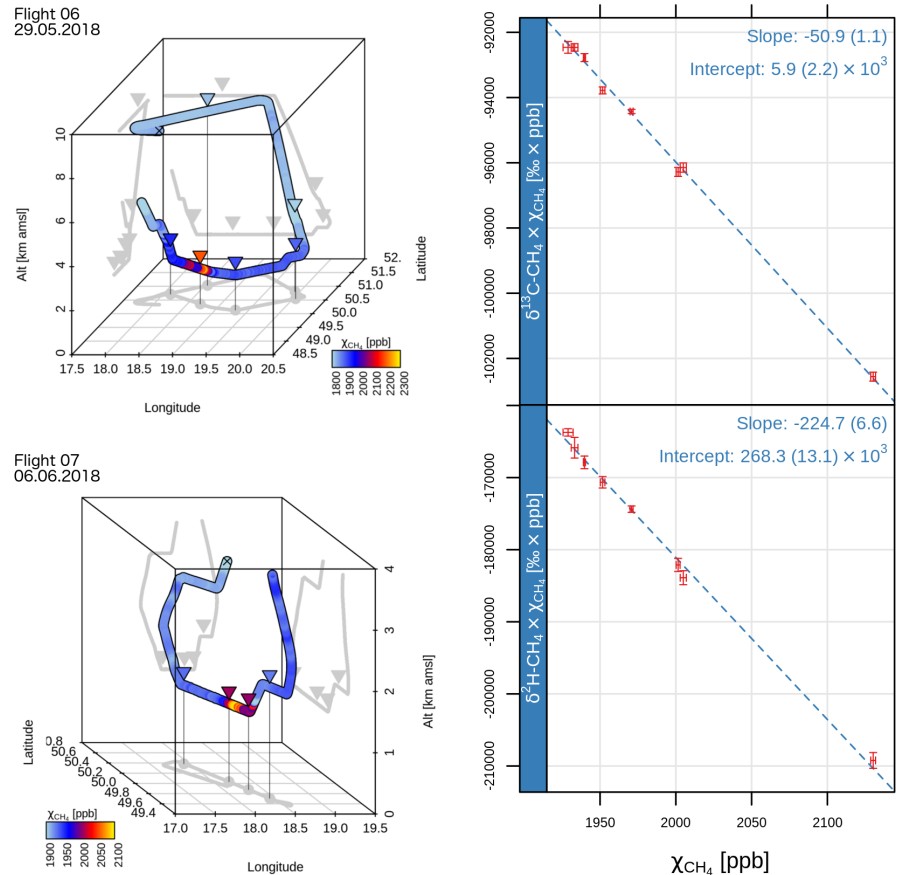

**Figure 8.** Left: Visualisation of $CH_4$ measurements over USCB during flights no. 6 (May 29[th], top panel) and no. 7 (June 6[th], bottom panel). Coloured lines represent mixing ratios along the flight path. Triangles show the mean flask sampling point, coloured according to the flask $CH_4$ mixing ratio. Both in situ and flask mixing ratios are coloured using the same scale. Right: Miller-Tans model of isotopic source signatures for $\delta^2 H$ and $\delta^{13} C$, based on flask samples collected over the USCB. For flight no. 7 only the four flask collected below 4 km were included in the analysis. See text description for details. The dashed line is the linear fit using the Williamson-York formula (Cantrell, 2008). Values of fit parameters are given with 1-$\sigma$ uncertainty in the parentheses.

7, respectively). The main effort of flask sampling over USCB was carried out using another platform, namely FDLR Cessna
(c.f. Fiehn et al., 2020, in review), aboard which a twin instrument was installed that allowed for a batch of approximately
60 samples to be collected, most of them inside the PBL. In the following paragraphs we only discuss samples collected on
HALO. A broader discussion of methane isotopic composition observed during CoMet 1.0 will be a part of an upcoming study.

During flight 6 (Fig. 8, top left), the aircraft crossed the USCB at approximately 8 km altitude, then turned south-eastwards
during the descent down to ca. 2 km, where it crossed into the PBL. Then, after turning south-westward, it crossed the USCB
area upwind of the coal-mine sources at 1800 m, and, after another turn to north-west, it crossed the study area again, this time



downwind of the known source locations. Along the flight path, 6 flasks were collected: two in the free troposphere during the high-altitude overpass and the descent, respectively, then two during the upwind crossing in the PBL and another two during the final downwind crossing inside the PBL. During flight 7 (Fig. 8, bottom left), a similar pattern was executed, with the exception that no upwind crossing in the PBL-section was executed. Again, two flasks were collected in the free troposphere immediately

before and during the descent (at altitudes of 7.6 km and 4.3 km a.m.s.l., not shown on the plot), and four subsequent samples were collected during the PBL-section, flown at an altitude of 1400 m for the most part, until the aircraft was forced to ascend to approx. 2 km after crossing into the airspace of Czechia.

A comparison between patterns of in situ mixing ratios and flask collection locations clearly shows that air masses with $CH_4$ mixing ratios significantly enhanced by local sources were sampled at least three times. We have aggregated measurements

from both days, assuming that the enhancement is coming from the same source (or source cluster) on both days, which is partially corroborated by wind observations on both days (not shown) and modelling analyses supporting the campaign (Nickl et al. and Gałkowski et al., in prep). Application of the Miller-Tans approach (Fig. 8A) yields an isotopic signature for the USCB source of $\delta^2H$ = -224.7 (6.6) ‰ and $\delta^{13}C$ = -50.9 (1.1) ‰. Again, standard uncertainty in the final digits is given in brackets.

These values can be compared to previously published data. A comprehensive data set, gathering published values of isotopic signatures from various methane sources has been compiled and described by Sherwood et al. (2017). Most of the information it contains came from studies focused on methane emitted from fossil fuel extraction (including regular oil drilling, shale gas extraction as well as on gas emitted during coal mining), but data on biomass burning and biogenic sources is also included. Figure 9 shows the main ranges of these source signatures together with flask data collected during CoMet 1.0. Fossil-related

methane sources are marked with rectangles (representing main ranges of the reported data and not their full extent) and biogenic source signatures as points (with bars marking the standard deviation of the reported signatures). As can be seen, the values reported in this study (marked with blue point with 1-$\sigma$ ellipse) fall into the typical range reported in Sherwood et al. (2017), in the middle of conventional anthropogenic methane sources characterised by relatively high $\delta^2H$ (in contrast to the second cluster with $\delta^2H$ values closer to -300 ‰).

We also compared our results against signatures from the same area published previously. Measurements of methane, with samples collected in the coal mine tunnels, were performed by Kotarba (2001). Their study encompassed measurements of methane isotopic composition from a total of 15 mines in the USCB (with $\delta^2H$ measured in all but one), including samples collected from different coal seams. For the purpose of this study we have aggregated distinct samples reported in the original study and calculated mine-specific averages, which yielded the total range of $\delta^2H$ between -202.0 ‰ and -157.5 ‰. Reported

values for $\delta^{13}C$ were between -77.1 ‰ and -44.5 ‰. These have been marked in Fig. 9 with a green rectangle labelled with "USCB".

In recent years there has also been a significant effort to constrain the USCB coal-mine source signatures, as many of the mines reported upon in Kotarba (2001) have been closed over the years, and those that remained open have completed the excavation of the old deposits and moved to different ones, possibly with different isotopic signatures. The results of these

more recent measurements, representing methane emitted from 23 different coal-mine shafts, have yielded mean values of the



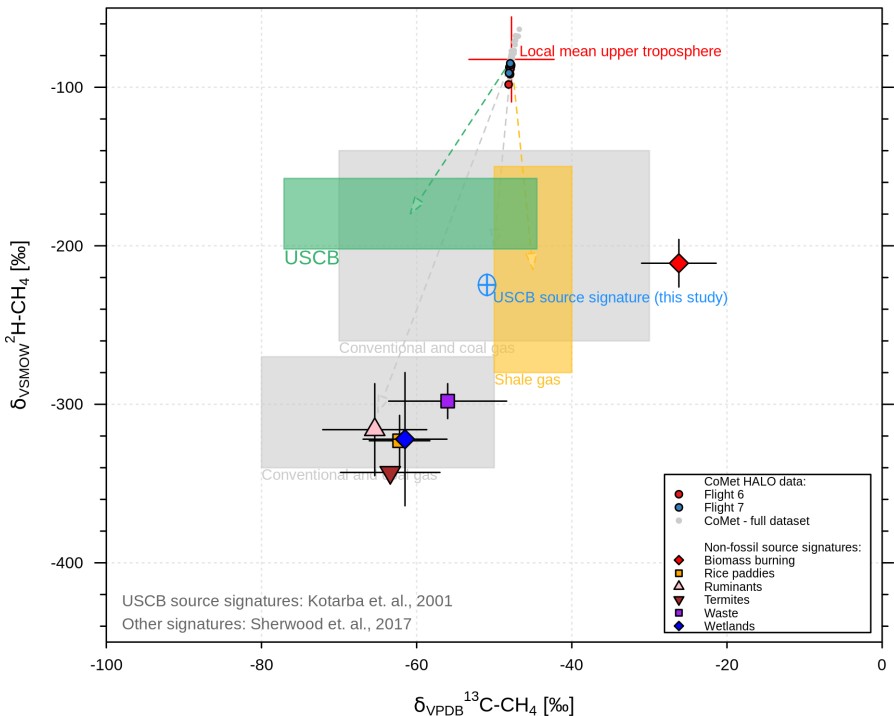

**Figure 9.** Isotopic composition of methane measured over the USCB (red and blue points) compared to source signatures estimated in this study using the Miller-Tans method (light blue cross with 1-$\sigma$ uncertainty ellipsis), and previously reported ranges of isotopic composition for the dominant methane source groups. For the Miller-Tans method, the background signature was estimated using available measurements from the free troposphere (red cross). Data on global source signatures is based on Sherwood et al. (2017). Coloured blocks denote the approximate broad ranges of values reported there for a) coal and conventional gas excavation (purple), shale gas excavation (orange) and biomass burning (blue). Selected biogenic source signatures are also marked as points, with whiskers denoting the standard deviation of values reported in Sherwood et al. (2017). Data reported previously for USCB coal-mine sources by Kotarba (2001) is marked with a green rectangle.

signatures ($\delta^{13}$C = -49.8 (5.7) ‰, $\delta^2$H = -206.1 (46.3) ‰, brackets showing standard deviation), indicating that the methane emitted there has shifted to lower deuterium signatures (with almost unchanged values of $\delta^{13}$C), corroborating the value reported in the present study. A more detailed discussion on these recent measurements will be presented in an upcoming study by Stanisavljevic et al. (in prep, this special issue).

**4    Conclusions**

A high-resolution in situ system for online observations of greenhouse gases (JIG) was successfully deployed during the CoMet 1.0 mission aboard the German research aircraft HALO aircraft over continental Europe. More than 55 hours of high frequency



(1 Hz) observations of $CO_2$ and $CH_4$, and over 38 hours of CO observations were collected over the course of 9 flights between May 15th and June 12th, 2018. In addition to in situ observations, 96 discrete flask samples were collected and analysed for

atmospheric composition, including $\delta^{13}C$ and $\delta^2H$ isotopic signatures of methane. Careful pre-flight, in-flight and post-flight calibration procedures allowed us to obtain a highly precise (single measurement standard deviations: 0.06 ppm - $CO_2$, 0.3 ppb - $CH_4$ and 3.1 ppb - CO) data record that is traceable to international WMO calibration scales. Comparison with flask samples analysed in the laboratory confirm that the measurement data comply with the WMO compatibility goals.

Observations collected during the mission were used in combination with two of the available modelling products (CAMS

and CarboScope) to explain the observed atmospheric variability on both regional scales as well as during the localised vertical soundings (a total of 50 throughout the campaign), covering altitudes from ground level to 14 km a.m.s.l.

Independent validation of available model products showed overall good agreement between observations and global state-of-the-art products, with very good agreement for $CH_4$ and $CO_2$ in the free troposphere / residual layer range (3–10 km) and slightly (CAMS) to significantly (CarboScope) worse performance in the PBL and the stratosphere. These results highlight i)

the inability of the coarse-grid models to represent local sources and processes influencing individual profiles (in particular for CarboScope, but also clearly visible in the relatively high resolution CAMS product), and ii) challenges in the high-resolution modelling of biospheric fluxes of $CO_2$.

We have also demonstrated the potential of using isotopic signatures measured in the downwind plume for source attribution. Samples collected during two flights above one of the main target areas of the CoMet 1.0 mission, the USCB, have clearly

pointed to coal-mining as the main source of the observed methane enhancement ($\delta^{13}C$ = -50.9 (1.1) ‰, $\delta^2H$ = -224.7 (6.6) ‰). It should be noted that while the measured deuterium signatures are substantially lighter than has been reported in previous studies from the area, they correspond to more direct estimates performed in the scope of CoMet 1.0 by other involved teams (M. Stanisavljevic, personal communication), highlighting a shift in isotopic emission signatures following changes in coal-mining activities, e.g. the closure of coal mines or changes of excavated coal-beds/seams.

*Code and data availability.* The code used for data processing and analysis is available from the first author per request. The data collected during the mission are available on the HALO database at halo-db.pa.op.dlr.de

*Author contributions.* MG collected the data and prepared the manuscript, with contributions from all co-authors. SB, MR and AJ assisted with flask sample collection and performed the analyses. AA-P supplied CAMS model data and assisted with data interpretation. JM, FTK and JC contributed to the mission planning and data analysis. CG co-designed the CoMet 1.0 mission, collected the data and provided critical

input to the manuscript.

*Competing interests.* The authors declare no competing interests.



*Acknowledgements.* We would like to express our appreciation to all the people involved in the CoMet campaign, without whom the campaign could not succeed. In particular we would like to thank DLR-FX for the campaign management, the pilots of the HALO aircraft, our colleagues at AGH University of Science and Technology and all the staff at MPI-BGC involved in the flask measurements. We also want
to thank Mila Stanisavljevic for providing information on the isotopic composition in USCB. We gratefully acknowledge funding for the CoMet 1.0 campaign by MPG (Max Planck Society) and by BMBF (German Federal Ministry of Education and Research) through project AIRSPACE (FK 390 01LK1701C). We also acknowledge the use of resources of Deutsches Klimarechnungszentrum (DKRZ), namely high-performance cluster Mistral, for data storage and analysis.





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
