# Peer review of "In situ observations of greenhouse gases over Europe during the CoMet 1.0 campaign aboard the HALO aircraft"

_Atmospheric Measurement Techniques, 2020_

## Referee Comment (RC1) · Anonymous Referee #1 · 17 Sep 2020

This paper presents a nice overview and synthesis of the two kinds of measurements made on the CoMet Aircraft, in-situ continuous and flask measurements, including isotopees. It also presents some analysis of the data, which in my view stretches a bit the goals for this journal, although I have definitely seen this type of thing before in AMT. The conclusion did a nice job of tying things together when really the results cover a lot of different topics, ranging from how well a global model reproduces vertical gradients to the isotopic value of the USCB. Very broad! Other than that, it is well-written so I have very few editorial remarks below, and a few requests for more explanation of some of the measurement techniques, such as if and how water vapor was removed for the in-situ system.

Details:

L69 ref to Varon is a satellite paper - this sentence reads as if it is a study using aircraft measurements.

L94 should read "instruments" L115 tolerance

L117 remove "so-called". I think working tanks is fine on its own.

L121, were these two cylinders at different values?

L 128, to be clear, the data itself was not adjusted for these in-flight calibration runs? Drift was assessed, was any drift found? If so, was it corrected? Why/why not? Were these in-flight calibrations noisier than expected so they were not used (see Karion et al., Long-Term Measurements of GHSs from Aircraft, AMT, 2013, for example)?

Was the sample dried prior to measurement by the Picarro? If so, does the calibration gas also pass through the drying system? If not, how was the effect of water vapor removed?

Somewhere in Sec 2 should be mentioned the quantity being measured, i.e. the dry air mole fraction of the species, with the definition that it is the moles of the species per mole of dry air, and define ppm as parts per million, or micromoles of $CO_2$ per mole of dry air... etc. These are formalities but they are useful so we keep the work accessible and clear.

L145-155, and throughout. Units should all be in metric, I see a lot of inches (") here. Inches I believe should be abbreviated as in. Perhaps give in cm with inches in parentheses?

L169- how was this drift discovered, was it by comparing the flask analyzed value from the lab with the in-situ system during flight? How big is "significant" (curious)?

L258: first time a broken mounting is mentioned, earlier it is referred to a roll-out malfunction. Perhaps either give a little detail or keep referring to it as a malfunction? I

think a sentence would be nice as to what happened exactly?

L289 and elsewhere, I would think approximately should be spelled out.

L300 and Section 3.4: I am starting to wonder if AMT is the appropriate forum for this extensive model-data comparison, as we are moving well beyond measurement techniques here.

Fig 8 These are impressive 3D renderings - this kind of data is difficult to visualize. But I am a little lost - if the plots on the right correspond with the flights on the left, then why are there more points in the Miller-Tans plots than on the left (i.e. the lower should only have 4 points then correct?). Something I am missing here?

---

## Referee Comment (RC2) · Anonymous Referee #2 · 22 Sep 2020

Review of "In situ observations of greenhouse gases over Europe during the CoMet 1.0 campaign aboard the HALO aircraft" – Galkowski et al.

**General Comments**

The manuscript presents a broad overview and analysis of results of aircraft in situ measurements (continuous greenhouse and trace gases) and atmospheric trace gas and isotopic composition measurements in onboard flask samples conducted during the CoMet 1.0 campaign. It details these two measurement systems and inter-comparisons in the context of WMO compatibility goals and presents model-data comparisons to two commonly used global modelling systems with different resolutions and meteorological drivers. The article is well-written, concise, and scientifically sound, but I feel that the focus of this manuscript stretches the limits of the scope of the AMT journal. As such, and since the article is part of a special issue collection, the majority of my specific comments aim to provide more detail for important measurement techniques below.

**Specific Comments**

Section 2.2.1 describes the CRDS measurements, but there is some key information and clarification lacking from the description of this system and its operation that should be included here. For example, air sample drying and/or water vapour corrections to these greenhouse gas measurements should be discussed as these measurements are compared to dried flask sample measurements. Furthermore, 'mixing ratios' measured should be reported as dry air mole fractions: please replace 'mixing ratio' with 'mole fraction' throughout the text. It would also be useful to also specify how many in-flight offset-corrections occurred on average per flight as these were performed manually, and specify whether this is a single-point correction or if two calibration tanks were used.

For flask samples, please clarify the level to which air samples are dried with the magnesium perchlorate and the pressure and volume of air that is sampled in each glass flask (is this 1 L at ambient pressures?). On which type of analyser were the flasks measured? L169 states that there was a drift in the CO flask measurements between collection and measurement; for clarity, please elaborate on how you have determined this using the in situ data. L194: Please also clarify how flasks were sampled during vertical profiles; if the aircraft was only either ascending/descending over the flask fill time, the reader can infer that the air samples collected during these profiles represents an integrative mole fraction between potentially several layers of the atmosphere. It might therefore also be useful to state, on average, how long it takes to fill a flask.

Throughout the manuscript, the Jena CarboScope model is referred to as TM3, but in L235, it is mentioned that it will be referred to as "CarboScope" – please choose one or the other for consistency.

It is mentioned in Section 4 that the flask measurements are compatible, by WMO standards, with the G2401 measurements but this does not seem to be the case. The in situ-flask differences, given uncertainties, fall outside of WMO surface compatibility goals for CO2 (0.1 ppm) and CH4 (2 ppb). As compatibility between measurement systems is mentioned as one of the scientific goals of the CoMet 1.0 mission, it seems necessary to state these compatibility goals for CO2 and CH4 as defined by WMO.

Figure 3 does not define the differences shown (i.e. are flask – in situ values shown)? I also wonder why these results are shown by flight number rather than something more informative for understanding differences between the two systems (e.g. difference vs. altitude or difference vs. mole fraction), separating flights 1-7 and 8-9. I would suggest a figure showing these differences as a function of either of these parameters (perhaps in SI) to shed light on why differences are seen between in situ and flask greenhouse gases.

L290: Please describe how it was evidenced that a stratospheric intrusion was crossed.

Figure 4, at times (L292, and elsewhere) is described in kilometers, but the figure itself is denoted in pressure altitude. The text would be more consistent with this figure if it were describing events in pressure levels as well.

Figure 5 shows modeled vertical profiles, which seems redundant with Figure 6. In addition, these model vertical profiles are not ever discussed. I would suggest removing these panels.

**Technical Corrections**

- L14: Uncertainties are given in parentheses and not quotes, please rephrase.
- L83: Please define 'HALO' if this is an acronym
- L101: G4201-m should be G2401-m, I believe.
- L101: 'fulfil' is missing an 'l'
- L115: Please change 'tollerance' to 'tolerance'
- L200: Please change to 'boundary'
- L222: Please eliminate "when possible" and "when not"
- L244: Change 'these' to 'and'
- L253: Change 'brackets' to 'parentheses'

Figure 2 caption: please define what red/blue shading means. Black crosses are hard to see – perhaps a thicker line would suffice. Altitude should be specified as km ASL.

L273-275 might be more easily understood if (a)-(d) were noted on Figure 4.

Figure 5 denotes CH4/CO/CO2 mole fractions as 'X[]', which is somewhat mis-leading as XCH4/XCO/XCO2 typically denote total-column mole fractions.

L369-370: The offsets in the 3-10 km range are actually responsible for the tail.

Figure 8 caption "For flight no. 7 on the four flasks..." (plural)

Figure S2: Please increase line width for purple crosses, as these are difficult to see.

---

## Author Comment (AC1) · 27 Nov 2020

**AMT-Discussions: Response to interactive review 1**

*Manuscript title: "In situ observations of greenhouse gases over Europe during the CoMet 1.0 campaign aboard the HALO aircraft", Gałkowski et al.*
*From: Anonymous Referee #1,*
*Review received and published: September 17th, 2020*

Note: Original reviewer's remarks are given as **bold and italicized text**. We've assigned each comment with a code for easier reference. Responses from authors are given below the comments.

***RC 1.1. This paper presents a nice overview and synthesis of the two kinds of measurements made on the CoMet Aircraft, in-situ continuous and flask measurements, including isotopes. It also presents some analysis of the data, which in my view stretches a bit the goals for this journal, although I have definitely seen this type of thing before in AMT. The conclusion did a nice job of tying things together when really the results cover a lot of different topics, ranging from how well a global model reproduces vertical gradients to the isotopic value of the USCB. Very broad!***

We thank the reviewer for this comment. Indeed, we spent quite some time thinking about the appropriate journal. We finally decided to put the measurement results at the core of our study. With this in mind, we thought that the demonstration of the usefulness of our measurements would fit our narrative, and comparison against widely-used state-of-the-art global models provided an excellent opportunity.

***Other than that, it is well-written so I have very few editorial remarks below, and a few requests for more explanation of some of the measurement techniques, such as if and how water vapor was removed for the in-situ system.***

***Details:***
***RC 1.2. L69 ref to Varon is a satellite paper - this sentence reads as if it is a study using aircraft measurements.***

We have moved the sentence in question to a paragraph discussing remote sensing techniques (after the text in L59).

***RC 1.3. L94 should read "instruments"***

Corrected.

***RC 1.4. L115 tolerance***

Corrected.

***RC 1.5. L117 remove "so-called". I think working tanks is fine on its own.***

Agreed.

**RC 1.6. L121, were these two cylinders at different values?**

Indeed, they were. We've added the information on the mole fractions of the calibration mixtures. Additionally, in L122, a minor correction regarding the length of calibration cycles was introduced. The respective fragment now reads:

"The instrument calibration was monitored during the mission with the use of two reference in-flight cylinders that contained dry mixtures of atmospheric air of known composition, for each tracer at a high and a low mole fraction, namely 373.4 -- 397.4 $\mu$mol mol$^{-1}$ for $CO_2$, 1661.0 -- 1917.1 nmol mol$^{-1}$ for $CH_4$, and 77.4 -- 139.5 nmol mol$^{-1}$ for CO. These were analysed several times during each flight. The calibration cycle consisted of two intervals, each three minutes in length. The first minute of each interval was discarded in subsequent analyses due to pressure equilibration effects within the regulators."

**RC 1.7. L 128, to be clear, the data itself was not adjusted for these in-flight calibration runs? Drift was assessed, was any drift found? If so, was it corrected? Why/why not? Were these in-flight calibrations noisier than expected so they were not used (see Karion et al., Long-Term Measurements of GHSs from Aircraft, AMT, 2013, for example)?**

Indeed, in-flight calibrations were used exclusively to monitor for drifts. No significant drift was found. The flight-to-flight variation of each low- and high-span measurement during the period prior to the instrument malfunction on June 7[th] was slightly larger than expected for $CO_2$, but not for other gases.

We added the following paragraph to the result section (3.1, approx.. L266 in the revised manuscript) of the manuscript, describing this in more detail:

"Results from in-flight measurements of the two reference cylinders showed no significant drift, however the flight-to-flight variation of each low- and high-span measurement during the period prior to the instrument malfunction on June 7[th] was slightly larger than expected for $CO_2$: low-span measurements varied by 0.10 $\mu$mol mol$^{-1}$, 0.4 nmol mol$^{-1}$, and 1.0 nmol mol$^{-1}$, while high span measurements varied by 0.14 $\mu$mol mol$^{-1}$, 0.3 nmol mol$^{-1}$, and 0.8 nmol mol$^{-1}$ for $CO_2$, $CH_4$ and CO, respectively. The likely cause for this are the silicon rubber membranes used in the pressure regulators (Filges, 2015), which are known to cause diffusion of $CO_2$ (Hughes, 1995). Given that species other than $CO_2$ did not show unexpected behaviour, we did not apply any correction of the measurements resulting from the in-flight measurements of the reference cylinders. For this reason, we also did not apply any correction of drift within each flight, in contrast to the experience of Karion et al. (2013b).

**RC 1.8. Was the sample dried prior to measurement by the Picarro? If so, does the calibration gas also pass through the drying system? If not, how was the effect of water vapor removed?**

The sample was not dried externally. The water correction was performed based on online measurements of $H_2O$ and followed the procedure described by Filges et al. (2015), which is

consistent with a more recent study from Reum et al. (2019). We have added this information in section 2.2.1. (after L118 in the revised manuscript). It reads:

"The instrument reports dry mole fractions, defined as number of molecules of each species in moles per one mole of dry air, with typical observed ranges expressed in µmol mol$^{-1}$ for $CO_2$ (equal to one part per million, ppm) and in nmol mol$^{-1}$ for CO and $CH_4$ (equal to 1 part per billion, ppb). As the collected air was not dried in the sampling line, a water correction was applied based on the online measurements of $H_2O$ mole fraction, following the approach described in previous studies of Filges et al. (2015) and Reum et al. (2019).

**RC 1.9. Somewhere in Sec 2 should be mentioned the quantity being measured, i.e. the dry air mole fraction of the species, with the definition that it is the moles of the species per mole of dry air, and define ppm as parts per million, or micromoles of CO2 per mole of dry air... etc. These are formalities but they are useful so we keep the work accessible and clear.**

We have added the short definition in subsection 2.2.1. together with the information on water correction (c.f. RC 1.8. above).

**RC 1.10.         L145-155, and throughout. Units should all be in metric, I see a lot of inches (")
here. Inches I believe should be abbreviated as in. Perhaps give in cm with inches in parentheses?**

Agreed, now OD given in mm with inches in parentheses.

**RC 1.11.         L169- how was this drift discovered, was it by comparing the flask analysed value from the lab with the in-situ system during flight? How big is "significant" (curious)?**

The first occurrence of the issue was observed when comparing the flask values against corresponding in situ observations done with JIG. Mean bias between these two was equal to -9.4 $\pm$ 1.2 nmol mol$^{-1}$ for flights 1—7 and -11.0 $\pm$ 2.6 nmol mol$^{-1}$ for flights 8—9, much larger than expected. For several flask samples analysis repetitions were made after a few days that yielded systematically higher CO results (1.5 – 3 nmol mol$^{-1}$ within 2 – 20 days indicating drift rates of up to 1 nmol mol$^{-1}$ d$^{-1}$.

To find the reason for that discrepancy, we have analysed flasks that were not filled during the campaign, which still contained the conditioned air samples from Jena. As the GasLab at MPI-BGC carefully maintains the mole fractions of these conditioning mixtures, we were able to diagnose that indeed the CO values of these flasks were drifting.

Following on these initial results, a lab experiment was performed in early 2019, where 10 flasks were equipped with different configurations of the sealing caps. One of the flasks tested was equipped with the same type of sealing caps as the flask set used during CoMet (namely 7 PCTFE type). It was found that for this type of cap the drift in CO was significantly larger than for those of the regular flask pool used in other field measurements supported by GasLab in Jena (about 2.9 nmol mol$^{-1}$ month$^{-1}$ per cap). No extra effect was observed for other gases.

As the exact reason behind this extra drift couldn't be established, and a precise correction function could not be calculated, we have decided to discard the CO measurements.

We have added some of the extra information to the manuscript in section 2.2.2:

"A significant (approximately 10 nmol mol$^{-1}$) bias in CO mole fractions was observed when comparing in situ measurements from JIG against gas flasks collected using JAS. Control laboratory experiments run after the campaign have shown that this bias was a result of a growth in CO mole fractions in the period between sample collection and subsequent laboratory analysis. This enhancement of the mole fraction could be attributed to new valve sealing polymer but could not be accurately corrected, therefore we have decided to discard these results. Careful quality control and additional tests did not show any sign of other gases being affected."

***RC 1.12.        L258: first time a broken mounting is mentioned, earlier it is referred to a roll-out malfunction. Perhaps either give a little detail or keep referring to it as a malfunction? I think a sentence would be nice as to what happened exactly?***

Thank you for pointing this out. Detailed description of the malfunctions is given in L134-141. The text (L288 in the revised manuscript) has been clarified and now reads:

"After the malfunction (see section 2.2.1.), i.e. for flights no. 8 and 9, these mean offsets were equal to 0.127 (68) $\mu$mol mol$^{-1}$ and -0.64 (91) nmol mol$^{-1}$ for the respective gases. While the difference of values as compared to flights 1--7 is statistically significant, it is still close to the WMO compatibility goal.

***RC 1.13.        L289 and elsewhere, I would think approximately should be spelled out.***

Agreed.

***RC 1.14.        L300 and Section 3.4: I am starting to wonder if AMT is the appropriate forum for his extensive model-data comparison, as we are moving well beyond measurement techniques here.***

We thank the reviewer for this remark. Partial answer to that question has been given already in our comment to the initial statement (RC 1.1.).

We would like to further underline that our intention in sections 3.2 – 3.4 was primarily to use the models to assist in interpreting the collected measurements. Thanks to that, we believe we can simultaneously:

a) Further increase the confidence in our measurement results (e.g. when observing positive or negative peaks in both the model and observations we can with more certainty assume that this is due to large– or regional–scale physical processes rather than equipment issues or local effects)
b) Better understand the causes behind the observed signals which would then allow us and scientific community in general to improve the measurement strategies for the future campaigns.

These goals can only be achieved if we can trust in the model results, hence a more detailed discussion was necessary. Perhaps the model comparisons included in sections 3.3 and 3.4 might not directly be related to measurement techniques, but we have decided to use this opportunity and expand the discussion, as we believe a mutual benefit for both modelling and observational communities could be thus achieved.

**RC 1.15.** *Fig 8 These are impressive 3D renderings - this kind of data is difficult to visualize. But I am a little lost - if the plots on the right correspond with the flights on the left, then why are there more points in the Miller-Tans plots than on the left (i.e. the lower should only have 4 points then correct?). Something I am missing here?*

Thank you for that comment. Miller-Tans plot on the right is done using a combined sample set from both flights shown on the left. We have excluded samples collected in higher layers of the atmosphere (above 3 km altitude), as we assumed that they represent an airmass of different origin due to large-scale transport phenomena. Based on the wind analyses and supporting modelling results we assume that we can aggregate both sample sets and treat them as representative of a mean source from the USCB. This is stated in L424-427. Two panels on the Miller-Tans plot are not plotted separately for two measurement days, but present $\delta^2H$ and $\delta^{13}C$ values measured in the combined dataset, i.e. each panel contains eight observation.

We have modified the caption of Fig. 8 for further clarification, and also added "data combined" in the panel label We have also corrected a minor mistake in the data subset description: the altitude threshold for flasks used in Miller-Tans plots was previously given as 4 km; the correct value is 3 km.

The new caption now reads:

"Left: Visualisation of $CH_4$ measurements over USCB during flights no. 6 (a) and no. 7 (b). For flight no. 7, only data from below 4 km altitude is plotted for clarity. Coloured lines represent mole fractions along the flight path, with the first plotted measurement marked with 'x', triangles show the flask sampling locations. Both in situ and flask mole fractions are coloured using the same scale. c) Miller-Tans model of isotopic source signatures for $\delta^2H$ and $\delta^{13}C$, based on eight flask samples collected below 3 km over the USCB during flights no. 6 and 7 together. See text description for details. The dashed line is the linear fit calculated using the Williamson-York formula (Cantrell, 2008). Values of fit parameters are given with 1-$\sigma$ uncertainty in the parentheses".

*References added to the revised manuscript:*

Karion, A., Sweeney, C., Wolter, S., Newberger, T., Chen, H., Andrews, A., Kofler, J., Neff, D., and Tans, P.: Long-term greenhouse gas measurements from aircraft, Atmospheric Measurement Techniques, 6, 511–526, https://doi.org/10.5194/amt-6-511-2013,

Reum, F., Gerbig, C., Lavrič, J.V., Rella, C.W., Göckede, M., 2019. Correcting atmospheric CO2 and CH4 mole fractions obtained with Picarro analyzers for sensitivity of cavity pressure to water vapor. Atmos. Meas. Tech. 12, 1013–1027. doi:10.5194/amt-12-1013-2019

---

## Author Comment (AC2) · 27 Nov 2020

**AMT-Discussions: Response to interactive review 2**

*Manuscript title: "In situ observations of greenhouse gases over Europe during the CoMet 1.0 campaign aboard the HALO aircraft", Gałkowski et al.*
*From: Anonymous Referee #2,*
*Review received and published: September 22nd, 2020*

Note: Original reviewer's remarks are given as **_bold and italicized text_**. We've assigned each comment with a code for easier reference. Responses from authors are given below the comments.

**General Comments**

**RC 2.1.** ***The manuscript presents a broad overview and analysis of results of aircraft in situ measurements (continuous greenhouse and trace gases) and atmospheric trace gas and isotopic composition measurements in onboard flask samples conducted during the CoMet 1.0 campaign. It details these two measurement systems and inter-comparisons in the context of WMO compatibility goals and presents model-data comparisons to two commonly used global modelling systems with different resolutions and meteorological drivers. The article is well written, concise, and scientifically sound, but I feel that the focus of this manuscript stretches the limits of the scope of the AMT journal. As such, and since the article is part of a special issue collection, the majority of my specific comments aim to provide more detail for important measurement techniques below.***

We thank the reviewer for that comment. As explained in our reply to reviewer #1, we are aware that some sections of our manuscript might be on the edge of the scope for AMT. We believe, however, that including some analysis of the measurements is beneficial for our manuscript and improves the overall value of the paper for the general scientific community.

We kindly refer the referees to our previous discussion of above (see RC 1.1. and RC 1.14.).

**Specific Comments**

**RC 2.2.** ***Section 2.2.1 describes the CRDS measurements, but there is some key information and clarification lacking from the description of this system and its operation that should be included here. For example, air sample drying and/or water vapour corrections to these greenhouse gas measurements should be discussed as these measurements are compared to dried flask sample measurements.***

We kindly refer to section RC 1.8. in this document.

**RC 2.3.** ***Furthermore, 'mixing ratios' measured should be reported as dry air mole fractions: please replace 'mixing ratio' with 'mole fraction' throughout the text.***

Agreed, units have been replaced as requested. See also our discussion in RC 1.9.

***RC 2.4. It would also be useful to also specify how many in-flight offset-corrections occurred on average per flight as these were performed manually, and specify whether this is a single-point correction or if two calibration tanks were used.***

We thank the reviewer for these remarks. We have taken them into the account when addressing point RC 1.7. See discussion there.

***RC 2.5. For flask samples, please clarify the level to which air samples are dried with the magnesium perchlorate and the pressure and volume of air that is sampled in each glass flask (is this 1 L at ambient pressures?).***

The air is dried to dew points below -70 °C. With regard to the pressure inside flask, the following text was modified in section 2.2.2. at approx. L159:

"(…) that provides the over-pressure necessary to flush and pressurise the flasks, up to approximately 1500 hPa."

***RC 2.6. On which type of analyser were the flasks measured?***

The description of the measurement instruments was expanded in section 2.2.2.:

"Gas chromatographic analysis of air in glass flasks is made with a gas chromatographic system based on two GCs (6890A, Agilent Technologies) equipped with a flame ionisation detector and a Nickel $CO_2$ converter (FID) for $CH_4$ and $CO_2$, an electron capture detector (ECD) for $N_2O$ and $SF_6$, a helium ionisation pulsed discharge detector (D-3-I-HP, Valco Instruments Co. Inc.) for $H_2$, and a HgO Reduction Gas Analyser (RGA3, Trace Analytical) for $H_2$ and CO. Additional analyses of $O_2/N_2$, $Ar/N_2$ and isotopic composition of methane $\delta^{13}C$ -$CH_4$ and $\delta^2H$ -$CH_4$ were carried out in the IsoLab of MPI-BGC (Sperlich et al., 2016). The typical measurement precision of the laboratory analyses is given in Table 1."

***RC 2.7. L169 states that there was a drift in the CO flask measurements between collection and measurement; for clarity, please elaborate on how you have determined this using the in situ data.***

Description has been expanded following the discussion of RC 1.11 here – we kindly refer to the discussion above.

***RC 2.8. L194: Please also clarify how flasks were sampled during vertical profiles; if the aircraft was only either ascending/descending over the flask fill time, the reader can infer that the air samples collected during these profiles represents an integrative mole fraction between potentially several layers of the atmosphere. It might therefore also be useful to state, on average, how long it takes to fill a flask.***

Indeed, the filling time was variable with height. The altitude to which the flask was assigned to was assigned based on the box model that took into the account flow, pressure and tubing volumes, as

described in L159-L163, following the work of Chen et al. 2012. We've added the following additional information into the text after 170:

"Each flask was flushed with 10 times its volume prior to closing the upstream and downstream valves. Typically, flasks were filled during descending profiles, but on some occasions also during ascents. The variable ambient pressure caused the flask fill time to vary between 100 seconds at high altitudes to 25 seconds close to the surface."

***RC 2.9. Throughout the manuscript, the Jena CarboScope model is referred to as TM3, but in L235, it is mentioned that it will be referred to as "CarboScope" – please choose one or the other for consistency.***

The plots and captions have been updated and now use CSc in place of TM3.

***RC 2.10.***       ***It is mentioned in Section 4 that the flask measurements are compatible, by WMO standards, with the G2401 measurements but this does not seem to be the case. The in situ-flask differences, given uncertainties, fall outside of WMO surface compatibility goals for CO2 (0.1 ppm) and CH4 (2 ppb). As compatibility between measurement systems is mentioned as one of the scientific goals of the CoMet 1.0 mission, it seems necessary to state these compatibility goals for CO2 and CH4 as defined by WMO.***

Thank you for this remark. In line 257 we have stated that the results are the difference is "still close to the WMO compatibility goal". However, the later statement in the "Conclusions" section was incorrect. This has been changed into (L508):

"Comparison with flask samples analysed in the laboratory confirm that the measurement data are close to compliance with the WMO compatibility goals (average bias smaller than 0.15 $\mu$mol mol$^{-1}$ and 3 nmol mol$^{-1}$ for $CO_2$ and $CH_4$, respectively)"

We have added the requested information on WMO compatibility goals in L285:

"The comparison between flask and in situ measurements is available for all except one flight (no. 5). From the 96 samples collected and analysed, 84 had simultaneous in situ measurements available from JIG that could be used for a bias assessment. Here, we compare bias between both our datasets to the 'network compatibility goal', defined by World Meteorological Organization as "the scientifically-determined maximum bias among monitoring programmes that can be included without significantly influencing fluxes inferred from observations with models" (WMO, 2019). WMO specifies this compatibility goal as equal to 0.1 $\mu$mol mol$^{-1}$ for $CO_2$ (in the northern hemisphere) and 2 nmol mol$^{-1}$ for both $CH_4$ and CO."

***RC 2.11.***       ***Figure 3 does not define the differences shown (i.e. are flask – in situ values shown)? I also wonder why these results are shown by flight number rather than something more informative for understanding differences between the two systems (e.g. difference vs. altitude or difference vs. mole fraction), separating flights 1-7 and 8-9. I would suggest a figure showing these differences as a function of either of these parameters (perhaps in SI) to shed light on why differences are seen between in situ and flask greenhouse gases.***

In the course of data analysis, we have investigated the mismatch between JIG and JAS in detail and found no clear relationship of in situ vs. flask mole fractions with either altitude or mole fraction (see figures below).

For both $CO_2$ and $CH_4$, the distribution of difference seems to be largely independent of altitude (Fig. A). Three outliers with very negative bias for methane (below 10 nmol $mol^{-1}$) are observed around 10 km, however even for high altitudes most of the differences are in the typical reported range (approximately -5 to 0 nmol $mol^{-1}$, with the mean of -2.93, nmol $mol^{-1}$).

As shown in Fig. B., the variability of JIG-JAS difference seems to be slightly higher closer to the higher end of the observed mole fraction range, however no clear trend can be observed. For methane, small biases are observed in the lower mole fraction ranges (below 1850 nmol $mol^{-1}$) with outliers for very low values of methane seemingly less biased than the ones observed in the typical tropospheric range (1900-1950 nmol $mol^{-1}$).

[Figure]

Figure A. Difference between JIG in situ and flask values, and their dependence on altitude, presented separately for flights 1—7 (red) and 8—9 (blue). Notches added to X and Y axes to display distribution of values (these are not colour-labeled).

[Figure]

*Figure B. Difference between JIG in situ and flask values plotted as a function of mole fractions. Notches were added to X and Y axes to display the distribution of values (notches are not colour-labeled).*

We've therefore decided that we want to emphasize first and foremost the change in the mean offset in flights 8—9, and to present the day-to-day variability, which is done in Figure 3 of the manuscript.

The plot has been updated in the following manner: i) difference is now clearly defined with the Y label axis; ii) units changed to $\mu mol\ mol^{-1}$ and $nmol\ mol^{-1}$ for $CO_2$ and $CH_4$, respectively, for consistency with the rest of the text (c.f. RC 2.3).

**RC 2.12. *L290: Please describe how it was evidenced that a stratospheric intrusion was crossed.**

'Stratospheric intrusion' in this sentence is meant to describe stratospheric air being pulled below typically observed altitude ranges. This is supported by comparison of CAMS model predictions and our in situ observations for all greenhouse gases (Fig. 4), where we see a clear segment of airmass with significantly reduced mole fractions, which is probably a potential vorticity (PV) filament that has been brought down by the outflow of the convective system. While we did not measure the ozone concentrations that are typically associated with stratospheric intrusions, and the observed air mass clearly doesn't reach deep into the troposphere, we believe that the very high correlation between observations and the model allows us to identify the phenomenon by stratospheric air presence at the flight level.

The text at L290 has been expanded, with stratospheric intrusion changed to 'stratospheric filament' in order to prevent confusion with the dry intrusions associated with large-scale tropopause folds:

"Immediately before the descent to Monte Cimone it crossed a stratospheric air filament, possibly brought down to the flight level by the outflow of a deep convective system active in the area in the afternoon on that day. This is corroborated by CAMS model results, which show a clearly

defined air-mass structure, depleted in mole fractions for all the observed compounds, stretching from the stratosphere at 200 hPa down to approximately 400 hPa (corresponding to roughly 13 km and 8 km a.m.s.l., respectively)."

**RC 2.13.** **Figure 4, at times (L292, and elsewhere) is described in kilometers, but the figure itself is denoted in pressure altitude. The text would be more consistent with this figure if it were describing events in pressure levels as well.**

Agreed. The text has been updated to use pressure levels together with altitudes.

**RC 2.14.** **Figure 5 shows modeled vertical profiles, which seems redundant with Figure 6. In addition, these model vertical profiles are not ever discussed. I would suggest removing these panels.**

We respectfully disagree. We believe that showing modelled profiles also in Fig. 5. gives a one-look visual comparison as to the nature of model vs. observation difference that would require careful, simultaneous analysis of both figures by the reader. Additionally, it also allows one to identify the distinct differences between CarboScope and CAMS products at the profile locations.

**Technical Corrections:**

**RC 2.15.** **L14: Uncertainties are given in parentheses and not quotes, please rephrase.**

Corrected.

**RC 2.16.** **L83: Please define 'HALO' if this is an acronym**

Agreed. Also defined in the abstract.

**RC 2.17.** **L101: G4201-m should be G2401-m, I believe.**

2.17. Indeed. We thank the reviewer for pointing this out.

**RC 2.18.** **L101: 'fulfil' is missing ans 'l'**

'Fulfil' is an acceptable form, used primarily in British English. (Oxford Advanced Learner's Dictionary, 7th Edition). We have left it as is.

**RC 2.19.** **L115: Please change 'tollerance' to 'tolerance'**

Corrected.

**RC 2.20.** **L200: Please change to 'boundary'**

Corrected.

**RC 2.21.** **L222: Please eliminate "when possible" and "when not"**

Sentence now reads:

"In order to estimate the background signature, we have used measurements from air samples collected in the immediate vicinity of the target plume, either from the upwind air masses or from outside of the main plume."

***RC 2.22.        L244: Change 'these' to 'and'***

Modified sentence now reads:

"Data from 51 vertical profiles are available, out of which 21 have simultaneous flask measurements. They are listed in the supplement (Table S1)."

***RC 2.23.        L253: Change 'brackets' to 'parentheses'***

Corrected.

***RC 2.24.        Figure 2 caption: please define what red/blue shading means. Black crosses are hard to see – perhaps a thicker line would suffice. Altitude should be specified as km ASL.***

The caption was updated according to reviewer's suggestions. The plot was also updated, with the crosses now 20% thicker and 20% larger; a small black dot was added in the centre on the top of the observations (blue line) to exactly define the value for 12.06.2018, where some of the crosses are largely covered by the data series. The width of the blue line was reduced by 30% to further make the crosses more visible. Similar changes were applied to Figure S2.

***RC 2.25.        L273-275 might be more easily understood if (a)-(d) were noted on Figure 4.***

Agreed.

The plot has been updated in the following manner: i) markings for periods a—d were added to the bottom three panels of the plot; ii) units have been changed to mole fractions, consistent with the rest of the manuscript; iii) vertical coordinate label switched to hPa. Caption and manuscript have been adjusted accordingly.

***RC 2.26.        Figure 5 denotes CH4/CO/CO2 mole fractions as 'X[ ]', which is somewhat misleading as XCH4/XCO/XCO2 typically denote total-column mole fractions.***

We have used a Greek letter $\chi$, which has been used before (instead of c) to denote mole fractions. See e.g. Röckmann et al., 2016. Similar notation was also used in Karion et al., 2013. This has been left as is.

***RC 2.27.        L369-370: The offsets in the 3-10 km range are actually responsible for the tail.***

We have clarified the description:

"Interestingly, the distribution of the mismatch in this altitude range is a positively skewed Gaussian curve (Fig. 6, bottom-right panel), with the values in the main peak almost symmetric around 0 µmol mol$^{-1}$, and the mean offset in the 3 – 10km range driven by the values in the tail of the distribution."

**RC 2.28.** *Figure 8 caption "For flight no. 7 on the four flasks…" (plural)*

No longer relevant, as this caption has been modified following to RC 1.15.

**RC 2.29.** *Figure S2: Please increase line width for purple crosses, as these are difficult to see.*

Figure S2 was updated similar to figure 2 (see RC 2.24 above). The label "TM3" was also replaced with Carboscope / CSc for consistency. The caption was updated accordingly.

*References added to the revised manuscript:*

Sperlich, P., Uitslag, N. A. M., Richter, J. M., Rothe, M., Geilmann, H., van der Veen, C., Röckmann, T., Blunier, T., and Brand, W. A.: Development and evaluation of a suite of isotope reference gasesfor methane in air, Atmospheric Measurement Techniques, 9, 3717–3737, https://doi.org/10.5194/amt-9-3717-2016, https://www.atmos-meas-tech.net/9/3717/2016/, 2016